# Construction of Bone Hypoxic Microenvironment Based on Bone-on-a-Chip Platforms

**DOI:** 10.3390/ijms24086999

**Published:** 2023-04-10

**Authors:** Chen Li, Rong Zhao, Hui Yang, Li Ren

**Affiliations:** 1Key Laboratory of Flexible Electronics of Zhejiang Province, Ningbo Institute of Northwestern Polytechnical University, Ningbo 315103, China; 2Key Laboratory for Space Bioscience and Biotechnology, School of Life Sciences, Northwestern Polytechnical University, Xi’an 710072, China; kittyyh@nwpu.edu.cn

**Keywords:** bone marrow, hypoxic microenvironment, cell culturing, bone disease bone-on-a-chip platform

## Abstract

The normal physiological activities and functions of bone cells cannot be separated from the balance of the oxygenation level, and the physiological activities of bone cells are different under different oxygenation levels. At present, in vitro cell cultures are generally performed in a normoxic environment, and the partial pressure of oxygen of a conventional incubator is generally set at 141 mmHg (18.6%, close to the 20.1% oxygen in ambient air). This value is higher than the mean value of the oxygen partial pressure in human bone tissue. Additionally, the further away from the endosteal sinusoids, the lower the oxygen content. It follows that the construction of a hypoxic microenvironment is the key point of in vitro experimental investigation. However, current methods of cellular research cannot realize precise control of oxygenation levels at the microscale, and the development of microfluidic platforms can overcome the inherent limitations of these methods. In addition to discussing the characteristics of the hypoxic microenvironment in bone tissue, this review will discuss various methods of constructing oxygen gradients in vitro and measuring oxygen tension from the microscale based on microfluidic technology. This integration of advantages and disadvantages to perfect the experimental study will help us to study the physiological responses of cells under more physiological-relevant conditions and provide a new strategy for future research on various in vitro cell biomedicines.

## 1. Introduction

All organs of the human body are exposed to different oxygen tensions, and some organs are exposed to very low oxygen levels and different oxygen gradients. The changes of the bone marrow hypoxic microenvironment have a certain impact on human physiological conditions, and therefore oxygen tension and oxygen gradient need to be considered for cell cultures in vitro. Over the past few years, scientists have developed many tools and technologies to create various spatiotemporal gradients for cell research. However, the current methods of studying the physiological activities of cells under hypoxia cannot accurately control the level of oxygenation at the microscale.

The development of microfluidic platforms can overcome the inherent limitations of these methods. Microfluidic technology is one of the most promising technologies at present because it has strong controllability in both the space and time domains. 

In addition to discussing the characteristics of hypoxic microenvironment in bone tissue, this review will also discuss various methods of constructing oxygen gradients in vitro and measuring oxygen tension from the microscale based on microfluidic technology. We will review the research on the implementation of these methods and finally report the main opportunities and key challenges we have found in this field.

## 2. The Hypoxic Microenvironment of Bone

### 2.1. Overview of the Bone Hypoxic Microenvironment

#### 2.1.1. Hypoxic Microenvironment Characteristics of Bone

Oxygen, an important source of energy for cellular metabolism and maintenance of the biological activity of the body, is transported to the whole body through the blood. To maintain oxygen homeostasis [1], hypoxic signaling pathways have matured to promote oxygen delivery and cellular adaptation to hypoxia [2]. Thus, hypoxic signaling plays an important role in development, tissue homeostasis, and pathological conditions [3,4]. 

Bone is a highly vascularized tissue that, however, harbors an extremely hypoxic microenvironment, with different regions of bone tissue characterized by different oxygen levels and oxygen gradients [5,6,7]. 

In the bone marrow compartment, quantification of bone marrow PO_2_ at several intravascular locations using two-photon light lifetime microscopy, as well as mathematical modeling, yielded values ranging from 11.7 to 31.7 mmHg, with a mean of 20.4 mmHg (2.7%) (Figure 1A), indicating that the bone marrow cavity is a hypoxic environment [8,9,10].

On cortical bone, the use of high-resolution imaging to study blood vessels and partial pressure of oxygen has been hampered by its thickness. Therefore, the partial pressure of oxygen to approximate cortical bone with a mathematical model is approximately 4.2% (30 mm Hg). Similar to the periosteum, oxygen levels in cortical bone are distributed as a result of presumably low cell density and low oxygen consumption [13].

On cancellous bone, the hypoxic microenvironment of the sinus region is similar to that of trabecular bone. When it is assumed that oxygen diffuses throughout the bone in a similar way to water (with the same diffusion coefficient), oxygen levels are expected to be even lower (0.08–2.4% or 0.6–17 mm Hg moving from the trabecular bone surface to the inner layer of osteocytes) [3,14,15,16]. Because this scenario has been predicted by mathematical models, the exact partial pressure of oxygen in trabecular bone remains unknown; however, models suggest that osteocytes embedded in the core of trabecular bone may experience a hypoxic environment [17,18]. 

At the endosteum, because the vasculature is different, smaller vessels are located closer to the endosteal surface by measuring PO_2_ at different locations within the BM. Vasculature with a diameter of 15 mm showed a higher partial pressure of oxygen (22.7 mmHg, 3.0%) than those with a diameter of 0.15 mm (19.5 mmHg, 2.6%; *p* < 0.03). Therefore, when analyzing PO_2_ values at different distances from the endosteum, researchers found that the lowest PO_2_ was located in a region of 0.40 mm [3,19]. These measurements revealed a moderate PO_2_ gradient with distance from the endosteum [20,21,22]. 

#### 2.1.2. Causes of Bone Hypoxic Microenvironment Generation

Physiological hypoxia exists in many tissues of the human body, and a low-oxygen environment is necessary to maintain normal physiological functions, such as those of the liver and bone. The low-oxygen microenvironment of bone may be determined by cellular levels and oxygen consumption rates in specific regions of the skeleton, where hypoxic effects at regions of the sinusoidal vasculature may further determine regional oxygen gradients; in addition, blood in bone flows from the central arteries into the arterioles, capillaries, and finally into sinusoids, and is associated with a progressive decrease in blood velocity and oxygen delivery. It has been suggested that sinusoidal blood flow is approximately one tenth of arteriolar blood flow, so low oxygen tension in bone marrow may result from low oxygen levels in sinusoidal blood flow and high oxygen consumption by hematopoietic cells [1,4,23] because these hematopoietic stem cells in the bone marrow compete with each other for the scarce nutrition and oxygen supply in the capillaries close to them (Figure 1B) [24]. In summary, oxygen tension in bone is primarily controlled by cell density, oxygen consumption, and oxygenated blood supply [19,25].

### 2.2. Effect of Hypoxia on Bone Function

#### 2.2.1. How Do Cells Sense Hypoxia?

How cells sense and adapt to oxygen levels in the microenvironment was previously unknown. Key to this process is the hypoxia-inducible factor (HIF) (Figure 1A). HIF is a binding protein present in humans and other mammals. It comprises four subunits, including O_2_-regulated α subunits (HIF-1α, HIF-2α, HIF-3α) and a β subunit (HIF-1β, or ARNT). Its main function is to induce bound proteins in response to low oxygen or anoxia. Among them, HIF-1β expression is generally unaffected by oxygen, but the expression of α-subunits is dependent on oxygen supply and has the shortest half-life [25,26].

In a normal oxygen environment, the oxygen-dependent domain (ODD) of α subunits is hydroxylated by proline hydroxylase domain enzymes (PHDs, including PHD 1, PHD 2, and PHD 3). Among these enzymes, PHD2 plays an important physiological function and PHD1 and PHD3 play a role in mitochondrial energy metabolism and innate immune responses, respectively (Figure 1C) [12,27]. HIF-α is ubiquitinated by the ubiquitin enzyme von Hippel Lindau tumor suppressor protein, and degraded, leading to a decline in intracellular HIF levels, which shapes the different sensing states of the cell in response to oxygen under different oxygen content environments: (1)In a hypoxic environment, HIF-α levels are elevated, resulting in the nuclear β dimerization of subunits. HIF-1 is an HIF-1 β and HIF-1 α of a heterodimeric protein complex that, upon binding to RNA polymerase II and interaction with major transcriptional coactivators such as P300, promotes the transcription of target genes. At the same time, it affects the expression levels of erythropoietin, glucose transporters, glycolytic enzymes, vascular endothelial growth factor, and other proteins, which ultimately help to increase oxygen delivery or promote metabolic adaptation to hypoxia, thereby regulating many important physiological behaviors, such as cell metabolism, erythropoiesis, and local angiogenesis [23,28].(2)Under normoxic conditions, inhibition of the asparagine hydroxylase factor hypoxia-1 transactivates the HIF-α of the conserved asparagine residues hydroxylated and interacts with P300 to ultimately repress HIF-induced transcription. HIF-3 α is thought to work by interacting with HIF-1 α competitive binding to inhibit HIF-1 α and HIF-2 β function [29]. Among these enzymes, PHD2 exerts major physiological functions. HIF-α is ubiquitinated and degraded by the ubiquitin protease tumor suppressor protein, resulting in reduced intracellular HIF levels, and, at the same time, a hypoxic environment can cause HIF-α levels to become elevated and associated with those in the nucleus β subunits undergoing dimerization [30,31,32]. Thereby, we can explore therapies for related diseases, such as ischemia, cancer, diabetes, stroke, infection, wound healing, and heart failure, by targeting the oxygen-sensing pathway.

#### 2.2.2. Effect of Hypoxia on Bone Function

Hypoxia on cell differentiation

In the past few years, several authors have studied the effect of low oxygen concentrations on cell growth in different cell models, such as rat bone marrow-derived mesenchymal stem cells and mouse fibroblasts, respectively [33,34]. D’ippolito et al. [35] demonstrated that low oxygen concentrations (1%, 3%, 5%, and 10% O_2_) increase cell proliferation. At 3% O_2_, the cells have the most suitable growth environment. In addition, the low partial pressure of oxygen inhibits osteoblast differentiation, indicating that a low partial pressure environment is one of the essential conditions for cell culture, while illustrating that low partial pressure of oxygen is essential for maintaining mesenchymal stem cells in a poorly differentiated state. Moreover, Lennon et al. [34] reported that rat bone marrow stromal cells showed increased extracellular mineralization in cultures maintained at low PO_2_ compared to normoxic conditions. In addition, in several in vitro studies, low oxygen concentrations have been found to stimulate the differentiation process, inducing the cells to progress towards osteogenic, adipogenic, or chondrogenic lineages [36,37]. From this we can determine that cell differentiation and survival can be affected by different partial pressures of oxygen.

Hypoxia on skeleton development

The effects of hypoxic environments on bone metabolism and development are complex, mainly through HIFs and pathways related to osteoblasts and osteoclasts, and differ with time and oxygen concentration. Hypoxia can promote osteoblast generation and differentiation [38,39]. Short-term hypoxia may promote the early proliferation and differentiation of osteoblasts, while long-term hypoxia may inhibit their proliferation and differentiation. Under a hypoxic environment, HIF stimulates osteoclasts mainly through acidification, whereas it increases bone resorption capacity [15,26,40]. Hypoxia can lead to osteoclast formation and depends on the duration of hypoxia. Increased anaerobic metabolism and accumulation of acidic metabolites in hypoxia can cause mild acidification of the local microenvironment, and osteoclast activation is precisely dependent on extracellular acidification. Osteoclasts increase in number, volume, and bone resorption capacity, with the greatest effect occurring at 2% O_2_ [41,42].

Hypoxia simultaneously controls many activities of osteogenesis, including angiogenesis, bone repair, osteoblast function, metabolism, and the size and activity of osteoclasts. In particular, the HIF signaling pathway plays an important role in bone formation during fracture repair. Conditional deletion of HIF-1 or HIF-2 in osteoblasts results in a significant reduction in bone volume, whereas HIF-1 and HIF-2 are over-stabilized by the loss of VHL, leading to increased bone mass and thus changes in bone volume [43,44,45].

Bone remodeling

In addition to regulating organogenesis and development, hypoxia also plays an important role in maintaining tissue homeostasis. In mature bone tissue, the HIF signaling pathway is involved in the stabilization of the intraosseous environment. This pathway regulates osteoblast-to-osteoclast crosstalk and becomes an important mechanism in medical or pathological bone remodeling processes, such as odontoplasty and osteoporosis. In odontoplasty, driven by forces, mature bone undergoes coordinated tissue resorption and formation. HIF signaling is involved in the coordinated differentiation of osteoblasts and osteoclasts [46,47,48]. During orthodontic loading, pressure-induced local hypoxia contributes to the stabilization of HIF-1, up-regulates the expression of vascular endothelial growth factor and RANKL in osteoblasts, and promotes osteoclast differentiation [11,29,49]. RANKL from osteoblasts interacts with the osteoclast precursor RANK, promoting osteoclast activation, whereas OPG (osteoclast inhibitory factor or osteoprotegerin) inhibits osteoclast differentiation. HIF promotes osteoblast-mediated osteoclastogenesis in bone homeostasis by inducing OPG expression, which prevents the resorptive activity of osteoclasts [50,51,52].

In a hypoxic environment, the pathway involved in bone metabolism changes, affecting the maturation and differentiation of osteoblasts/osteoclasts. For osteoblasts, hypoxia mainly occurs at the early stage of their differentiation. Hypoxia promotes the early differentiation of osteoblasts and generates corresponding signals to stimulate the maturation and mineralization of the matrix [3,15,53,54]. Short-term hypoxia through up-regulation of HIF-1 α promotes matrix mineralization, promotes osteoblast differentiation and maturation, and accelerates osteogenesis. For osteoclasts, hypoxia can increase the production of osteoclasts, which has nothing to do with the differentiation stage of hypoxia, but the duration and severity of hypoxia may affect the differentiation of osteoclasts. During hypoxia, the anaerobic metabolism dominates and acid metabolites accumulate, leading to local mild acidosis. Existing studies have found that the severity and duration of tissue exposure to hypoxia and the stage of osteoblast differentiation in which hypoxia occurs may affect the growth and reconstruction of bone [5,55,56,57].

Osteoimmunology

The HIF signaling pathway enhances bone formation and bone immunity by regulating immune cell activity, important pathways, and important secretory molecules. Bone immunology includes the molecular and cellular interactions between osteoblasts, osteoclasts, lymphocytes, and hematopoietic cells [58,59,60]. 

In 2012, Dandajena et al. demonstrated that peripheral blood mononuclear cells co-cultured with osteoblasts can obtain functional osteoclasts by triggering HIF-mediated differentiation. In the pathogenesis of bone- or cartilage-erosive diseases, such as periodontitis and rheumatoid arthritis, HIF-1α plays an important role in synovitis and osteoclast formation of macrophages or monocytes. The formation of osteoclasts may be related to the stimulation of RANKL. Inhibition of HIF-1 α reduces osteoclast production by down-regulating the RANK/RANKL/OPG signaling pathway. The RANK/RANKL/OPG signaling pathway is one of the most classical signal pathways in bone immunology, and it plays an important role in immune organs and bone development [14,15,31,61]. 

The HIF-1 signaling pathway is closely related to the RANK/RANKL/OPG signaling pathway because it activates HIF-1 α and up-regulates the expression of RANK and RANKL. In addition, the involvement of the HIF-1 signaling pathway in bone immunology is believed to lead to B lymphocytes, IL-1 β, and S1P-S1PR1 signal pathway regulation [29,62,63].

#### 2.2.3. Hypoxia and Bone-Related Diseases

Tumor bone metastasis

The reason for hypoxia caused by solid tumors is blood flow obstruction [10,29,64]. Increasing amounts of evidence show that hypoxia enhances the malignant phenotype of cancer cells by promoting angiogenesis, epithelial–mesenchymal transition (EMT), invasion, tumor stem-cell-like phenotype, tumor cell dormancy, and release of extracellular vesicles (EV). Most of them are HIF-dependent. HIF also plays a major regulatory role in angiogenesis, which is important in tumor metastasis [11,29,65].

Repair of bone defect

Bone defect refers to the damage to the integrity of bone structure caused by local bone deficiency due to congenital malformation, trauma, inflammation, tumor, and iatrogenic surgical treatment. Generally speaking, defects of limited scale can repair themselves by osteogenesis. HIFs play a key role in promoting acquired pathological ectopic osteogenesis; there may be similar oxygen sensing and action mechanisms that affect the final osteogenic effect by affecting the key cell activity and cytokine secretion.

Additionally, hypoxia is an indispensable factor in inducing angiogenesis. At the cellular level, hypoxia inhibits ATP consumption during metabolism, resulting in abnormal cell behavior. HIFs can regulate the expression of many important genes, which are essential for cell survival, angiogenesis, glycolysis, proliferation, and migration [31,66]. HIF is also a key factor to control the function of immune cells and plays an important role in inflammatory reaction.

Osteoporosis

Osteoporosis occurs when the balance between bone resorption and bone formation is disrupted during bone metabolism, resulting in bone loss and altered bone tissue architecture. In this process, osteoblasts and osteoclasts play a particularly important role. Under the condition of hypoxia, the growth and differentiation of osteoblasts is delayed and the mineralization of the bone matrix is inhibited, thus limiting the overall bone formation. Osteoblasts respond to the hypoxic environment, release adenosine triphosphate (ATP), inhibit bone formation, and stimulate osteoclasts, resulting in an increase in the number and activity of osteoclasts [23].

Hypoxia and hypoxic pathway proteins are influential for stromal cells in the bone/bone marrow environment. They directly regulate bone homeostasis, causing osteolytic damage, leading to the destruction of the normal structure of bone, resorption and loss of bone mass, and ultimately the occurrence of osteoporosis.

## 3. Construction of Hypoxic Microenvironment Using Microfluidic Platform

### 3.1. Factors to Be Considered in Building Hypoxic Microfluidic Platforms

#### 3.1.1. Selection of Microfluidic Materials

In the manufacturing process of microfluidic devices, there are many materials to choose from.

Firstly, silicon materials were previously the main materials used in early microfluidic platforms. When they are used to build a low-oxygen microenvironment, they show outstanding advantages and include a good degree of finish and mature processing technology, high strength, moderate price, high purity, and corrosion resistance, and can be used for micro-pumps, micro-valves, molds, and other devices. However, at the same time, their disadvantages include the difficulty of deep etching, the bonding success rate of silicon substrate being low, and the poor light transmittance limiting many observation methods [67,68,69,70]. 

Secondly, the microfluidic paper platform is a new type of microfluidic platform developed in recent years. When it is used to build a hypoxic microenvironment, its outstanding advantages are good biocompatibility, low cost, simple post-treatment, and non-pollution. At the same time, its disadvantages include the difficulty of deep etching, low intensity, not being resistant to corrosion, etc. [26,71,72,73]. 

Finally, polymer materials are currently mainstream materials, and their outstanding advantages when used to build a low-oxygen microenvironment include rich varieties; convenient processing and molding; and good price, optical properties, chemical inertia, electrical insulation, and thermal properties. Among them, polydimethylsiloxane (PDMS) is widely used in the research and application of microfluidic devices due to their obvious advantages [74,75,76]. Because of the elasticity of this material, it can be better integrated with external components, has non-toxic and breathable properties, and can form a sufficiently stable temperature gradient for cell survival. Therefore, it shows an irreplaceable position compared with other polymer materials [77,78,79].

#### 3.1.2. The Necessity of Constructing Oxygen Spatiotemporal Gradient

In the bone tissue structure, it can be summarized from the literature data that the partial pressure of oxygen on the periosteum is approximately 22.7 mmHg, the partial pressure of oxygen on the cortical bone is approximately 4.2% (30 mm Hg), the oxygen level on the cancellous bone is expected to be lower (17–0.6 mm Hg), and the average value of the partial pressure of oxygen in the bone marrow cavity is approximately 20.4 mmHg [80].

Traditional experimental studies of hypoxia do not consider the gradient of hypoxia, so it is impossible to simulate the in vivo growth environment of different bone cells under the same oxygen partial pressure. Different bone cells do not grow in a suitable oxygen environment, and their function, amount, and activity will be affected. The constructed hypoxic microenvironment model is not accurate enough, thus affecting the experimental results. In vitro bone-on-a-chip platforms are expected to realize accurate reconstruction of the oxygen concentration and gradient observed in vivo [81].

However, in order to further improve and utilize the advantages of microfluidic perfusion technology over traditional static culture technology, it is necessary to combine sensors to monitor relevant parameters. As an important component of each cell’s metabolism, it is also a regulatory parameter in cells and tissues. Increased understanding of its function and precise control of its concentration may be beneficial to the activity and function of incubated tissues or cells. A well-defined oxygen gradient can be established based on microfluidics [82].

In addition to bone tissue, almost every metabolic process of the human body needs oxygen. Vascular transport, molecular diffusion, and metabolism will produce complex oxygen gradients in different organs, thus enabling complex symbiotic microbial communities and living tissues to coexist stably. Each organization and cell has its own viability and unique oxygen demand. The establishment of an appropriate oxygen gradient can make the cells stably cultured in vitro. Therefore, it is essential to cultivate organ and tissue models in vitro under physiological-related oxygen concentrations to ensure that they can best simulate their functions in vivo [80].

### 3.2. Construction of Hypoxic Microenvironment Using Microfluidic Platforms

#### 3.2.1. Construction of Hypoxic Microenvironment by Using Space-Constrained Chemical Reaction

When a chemical deoxidizer is used to reduce the oxygen content in the microfluidic device to establish a hypoxic microenvironment, oxygen will be consumed according to the inherent chemical composition of the deoxidizer [83]. 

The elastic material PDMS, which has high gas permeability and only allows gas exchange, is commonly used to construct microfluidic devices. As shown in Figure 2A, the PDMS device is designed as a single layer with three groups of channels; the middle channel (1 mm wide) is used to culture cells, and the two side channels (100 mm wide) are used to scavenge and generate oxygen, respectively. The cell culture channel and chemical reaction channel are separated by a thin PDMS wall with a thickness less than 50 μm. Due to the high diffusivity of oxygen in PDMS, oxygen can freely diffuse between the cell culture and chemical reaction channels. Thus, oxygen scavenging and generation reaction in the designated channels can be used to control the oxygen gradient in the cell culture channel. In order to effectively generate the oxygen gradient in the cell culture channel, chemical reactants are introduced into the channel from two separate entrances and begin to mix with each other in the mixing area before flowing into the chemical reaction channel [84].

The device can control the oxygen gradient without using huge, pressurized gas cylinders, adding deoxidizer directly, complicated gas interconnection, or complex flow control. The single-layer structure of the device makes the observation of cell microscopes more intuitive and the equipment manufacturing easier. In addition, the device uses space-constrained chemical reactions to generate or remove oxygen in the microfluidic channel of the cell culture without direct chemical contact.

#### 3.2.2. Construction of Hypoxic Microenvironment with Sodium Sulfite Low-Oxygen Layer

PDMS can also be used to construct multilayer microfluidic devices, in which an ultra-thin PDMS membrane (30 μm) is commonly used to separate the top layer and the bottom layer, as shown in Figure 2B. The configuration is used to culture cells in a well (the top layer) with hypoxic microenvironment (achieved by the bottom layer). The bottom layer consists of two independent cross-channels, which are used to respectively transport the oxygen scavenger and nitric oxide donor solution to the required location, thus a controlled hypoxic microenvironment can be achieved in the top cell culture well. 

The advantage of a two-layer system is that cells will not be exposed to scavenger/donor chemicals, and both treatments can be patterned separately, thus eliminating the possible chemical reaction between donor/absorption chemicals (Figure 2B). The simultaneous manipulation of multiple cellular gaseous microenvironments by microfluidic devices can be used for research that requires precise control of hypoxia and nitric oxide in cells, such as in the early stage of wound healing [85].

#### 3.2.3. Constructing Hybrid Hypoxic Microenvironments Using Polydimethylsiloxane Polycarbonate (PDMS-PC)

Due to the high gas permeability of PDMS, oxygen around the circumstances may influence the constructed hypoxic environment in the device. Polycarbonate (PC) film with non-gas permeability can be embedded into the device as a gas diffusion barrier, as shown in Figure 2C [67]. The device is composed of two PDMS layers: the top layer is used for space-constrained chemical reaction to generate oxygen gradient in the cell culture room, the bottom layer is used to generate chemical gradients in the downstream cell culture room. A PDMS membrane with a thickness of 100 μm is sandwiched between the two layers to prevent the chemical substances in the top layer from directly contacting with the cell culture medium. 

The equipment can be operated with a small amount of chemicals without large gas cylinders and complicated flow control schemes. In addition, it can be directly used in a traditional incubator with an injection pump to simplify the system setup. Additionally, the oxygen gradient can be designed using mature channels based on a serpentine channel layout. The result is a microfluidic cell culture device that produces stable vertical chemical and oxygen gradients at the same time. The device has long-term stability and good cell incubator compatibility [87].

#### 3.2.4. Establish Hypoxic Microenvironment by Hypoxia Gas

Hypoxia gas is also commonly used to construct hypoxia environments in PDMS microfluidic devices, as shown in Figure 2D [86]. The constructed device has a three-layered structure; from top to bottom these are the gas layer, the cell culture layer, and the glass substrate. These three layers are bonded together, providing a structure for the diffusion of gas–liquid. The cover structure and bottom structure are clamped with polymethylmethacrylate and fixed with screws to make a reconfigurable device. Oxygen-containing gas (0 and 20.9%) is supplied from the air inlet with air pressure and controlled by the pressure reducing valve. Then, a controlled hypoxic environment is constructed in the cell culture layer. An oxygen probe is used to measure indoor oxygen concentration in real time. 

The platform can accurately control the oxygen concentration in different cell culture areas. Combined with the oxygen sensing system, oxygen concentration can be monitored and adjusted in real time [86,88].

### 3.3. Oxygen Level Detection Methods in Microfluidic Devices

#### 3.3.1. Oxygen Sensitive Fluorescent Reagent

The inclusion of an oxygen sensitive reagent is the most commonly used strategy for oxygen concentration measurement in hypoxic microfluidic devices [89]. Ruthenium complexes (e.g., tris(2,2′-bipyridyl) dichlororuthenium(II) hexahydrate (Ru complex)) are well-characterized oxygen-sensitive fluorescent reagents; their fluorescence is quenched in response to oxygen [90]. The correlation between fluorescence quenching and oxygen concentrations is linear and can be plotted using the Stern–Volmer equation (Equation (1)) [90].
(1)$$\frac{I_{0}}{I}-1=K_{SV}[O_{2}]$$
where *I*_0_ is the fluorescence intensity under theoretical 0% oxygen, *I* is the fluorescence intensity at a specified [O_2_], and *K_SV_* is the Stern–Volmer coefficient, which is the fluorescence quenching constant.

To experimentally measure the oxygen concentrations in the microfluidic device, the Ru complex solution is usually perfused into the microfluidic channels or integrated into the microfluidic device layer as an oxygen sensor. The fluorescence intensity captured by fluorescence microscopy is negatively correlated with the oxygen concentrations in the microfluidic device, and the oxygen level can be calculated using the Stern–Volmer equation.

The advantages of the fluorescent-based oxygen sensing method are very prominent. Its result is not dependent on the flow rate in microfluidic channels, it works well in electromagnetic fields, is stable after calibration, and has a noninvasive readout. However, its photobleaching cannot be ignored, and it is interfered with by chlorine [91].

#### 3.3.2. Oxygen Electrode

The Clark electrode is another commonly used method for detecting oxygen concentrations in hypoxic microfluidic devices [91]. When a voltage is applied between the oxygen electrodes and exceeds the decomposition voltage of O_2_ (approximately 0.2 V), an electrolytic current is produced between the electrodes. When the applied polarization voltage reaches a certain value, the diffusion current depends entirely on the concentration of O_2_ in the tested solution. Thus, under the conditions of constant polarization voltage and temperature, the magnitude of diffusion current can be used as the basis for the quantitative determination of dissolved oxygen [92].

The Clark electrode-based oxygen sensing method has been well established, and the electrode can be easily sterilized and integrated into the microfluidic device for real-time oxygen detection [93]. However, consumption of O_2_ during measurement, the drift of readings over time, their dependence on hydrodynamic conditions, a more laborious maintenance procedure, and the interfering action of a number of substances (e.g., H_2_S) cannot be ignored [91].

## 4. Hypoxic Bone-on-a-Chip Models

### 4.1. Tumor Bone Metastasis

Bone metastasis of breast cancer is one aspect of tumor bone metastasis currently being studied by organ-on-a-chip platforms. 

The process of obtaining samples of bone metastasis from human patients is extremely complex, and the number of in vivo models that can effectively simulate human bone metastasis is limited. Therefore, the research on tumor bone metastasis is limited, and it is inconvenient to identify or real-time monitor cancer cells in vivo. Moreover, the existing in vitro models, which are commonly based on two-dimensional (2D) monolayer culture, cannot simulate the bone microenvironment or completely replicate the metastatic pathological state [64,94,95,96].

Marturano-Kruik et al. developed a bone-on-a-chip model to mimic the bone perivascular niche of breast cancer bone metastasis, and the breast cancer cell colonizing and drug resistance was investigated under low level oxygen gradients and a perfused microenvironment [97]. The oxygen gradients resulted from the diffusive and convective transport of oxygen in a 3D scaffold coupled to its consumption by cells, and the oxygen concentrations across the length of the 3D scaffold is dependent on interstitial velocity. In this hypoxic and interstitial flow microenvironment, breast cancer cells persist in a slow-proliferative state associated with increased drug resistance. Oxygen gradients regulate cancer metastasis progression, and modeling the early events of metastatic colonization using biomimetic microfluidic devices may be useful to identify the key players mediating cancer latency toward designing more effective therapeutic strategies [16,17].

Hao et al. reported a bone-on-a-chip model for the spontaneous growth of 3D, mineralized, collagenous bone tissue, and co-culture of metastatic breast cancer cells with the osteoblastic tissues [88]. PDMS was used to fabricate the device; this material allows free exchange of oxygen and carbon dioxide between cell culture and the environment due to its high gas permeability, and therefore the cultures were under atmospheric oxygen level. Although unique hallmarks of breast cancer bone colonization (e.g., cancer cell dormancy), previously confirmed only in vivo, were observed [88], it is still very important to study breast cancer bone metastasis in a physiological relevant model with a hypoxic microenvironment in the future. 

### 4.2. Osteoporosis

Osteoporosis is a chronic metabolic bone disease mainly caused by an imbalance in osteoblast and osteoclast activity [98]. The elderly population and postmenopausal women are at high risk of osteoporosis. Although many in vivo and in vitro models have been developed, due to the high complexity of the bone microenvironment, these research models are not ideal [99,100]. With the increase of the elderly population, it is necessary to develop more accurate biomimetic bone models for effective osteoporosis drug screening and accurate evaluation of personalized medicine [59,101,102]. 

Paek et al. developed a high-throughput biomimetic bone-on-a-chip platform to recapitulate the key matrical and structural aspects of the natural bone microenvironment (known as the osteon) using an osteoblast-derived decellularized extracellular matrix [103]. Based on the platform, the drug efficacy of anti-SOST antibody, which is a newly developed osteoporosis drug for bone formation, was tested. However, the in vivo hypoxic microenvironment was not recapitulated in this platform. It will be necessary to construct low oxygen level gradients for further new osteoporosis drug development due to the critical role of hypoxia in osteoblasts and osteoclast functions [103,104,105].

### 4.3. Bone-Marrow-on-a-Chip

Pluripotent hematopoietic stem cells and progenitor cells (HSPCs) are the source of all blood cell types. It is known that the bone marrow stem cell niche in which the HSPC are maintained is vital for their maintenance [106,107]. However, until now, no in vitro model has been presented that can truly simulate all aspects of the bone marrow niche and allow the long-term culture of HSPCs. Sieber et al. presented a novel 3D co-culture microfluidic model using a hydroxyapatite-coated zirconia scaffold, within which a molecular and structural microenvironment similar to the in vivo bone marrow niche was formed [106]. Based on this platform, long term HSPC culture and primitive state (CD34^+^CD38^−^) maintaining was realized; however, the low oxygen level gradients in the bone marrow [8,9,10] were not considered. Maintaining mesenchymal stem cells (MSCs) in a poorly differentiated state is also an important issue for stem cell therapy, and previous researches have demonstrated that low partial pressure of oxygen is essential for maintaining MSCs [35]; thus, constructing bone-marrow-on-a-chip with physiological hypoxia is necessary. 

## 5. Conclusions and Prospect

This paper introduces the influence of the hypoxic microenvironment on bone and the mechanism underlying this and summarizes various methods of constructing hypoxic microenvironments using microfluidic devices. At the same time, the hypoxic bone-on-a-chip platforms for the study of different bone-diseases were introduced. 

Biomimetic three-dimensional (3D) models based on an extracellular matrix synthesized through a biologically inspired process are commonly used for biological research due to their highly reproducible morphology and tunable macro- and micro-structure, and the hypoxic core is also formed because of the limited oxygen and nutrient diffusion through the 3D matrix [108]. However, the 3D models failed to emulate the complex chemical/oxygen gradients and interstitial fluidic microenvironment of the native bone. The microfluidic systems can be fabricated with minimum instruments, a small amount of reagents, and simplified syringe pump equipment, which makes the platforms more suitable for biomimetic bone biological research. The specially designed microfluidic parallel serpentine channels and gradient generators can successfully generate stable vertical chemical and oxygen gradients for cell research [73], which may help biologists to better study cell responses under a gradient combination that exists in the physiological microenvironment. In addition, different fluid shear requirements can be realized in the microfluidic platforms. Microfluidic devices provide more biomimetic bone tissue culture conditions and have shown great prospects and advantages in the research of bone biology and bone disease in vitro for various biomedical applications.

## Figures and Tables

**Figure 1 ijms-24-06999-f001:**
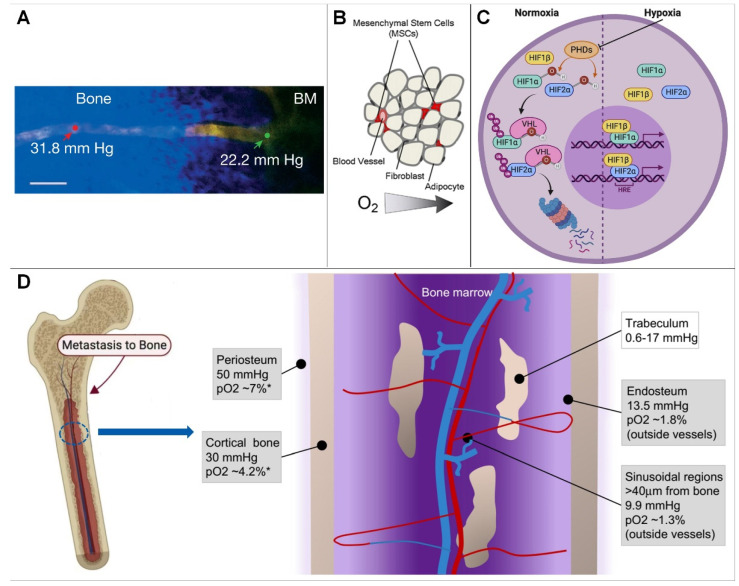
(**A**) Blood vessel projection image from bone to bone marrow (Reprinted with permission from Ref. [2], Copyright 2014, Springer Nature). (**B**) In the hematopoietic stem cell niche, cells compete for scarce nutrients and oxygen (Reprinted with permission from Ref. [1], Copyright 2010, Elsevier). (**C**) HIF signaling pathway (Reprinted with permission from Ref. [11], Copyright 2020, Elsevier). (**D**) Hypoxic microenvironment in bone marrow. * = Estimated conversion to pO_2_ from mm Hg(Reprinted with permission from Refs. [5,12], Copyright 2017, Elsevier).

**Figure 2 ijms-24-06999-f002:**
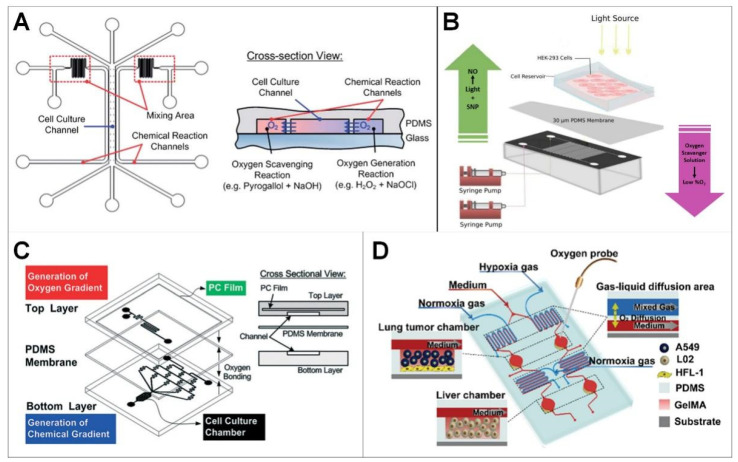
Construction of hypoxic microenvironment using microfluidic platform. (**A**) Microfluidic device capable of generating oxygen gradients using spatially confined chemical reactions (Reprinted with permission from Ref. [84], Copyright 2011, Royal Society of Chemistry); (**B**) Construction of hypoxic microenvironment with sodium sulfite low-oxygen layer [85]; (**C**) Constructing hybrid hypoxic microenvironments using polydimethylsiloxane polycarbonate (PDMS-PC) (Reprinted with permission from Ref. [67], Copyright 2014, Royal Society of Chemistry); (**D**) Construction of hypoxic microenvironment by hypoxia gas (Reprinted with permission from Ref. [86], Copyright 2021, American Chemical Society).

## Data Availability

Not applicable.

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
