# Peer review of "Construction of Bone Hypoxic Microenvironment Based on Bone-on-a-Chip Platforms"

_ijms, 2023, doi:10.3390/ijms24086999_

Round 1

Reviewer 1 Report

In this review, Li C. et al., attempt to discuss the advantages and disadvantages of various methods to construct oxygen gradients in bone-on-chip platforms. Meanwhile it can be considered an interesting topic, the review needs some improvements in order to be feasible for publication in the Journal.

Major comments:

The writing needs to be improved. Different paragraphs are difficult to understand because of the wrong syntax and grammar

Authors should add a first paragraph below the introduction about the biology of bone environment in order to contextualize how the hypoxia can affect cell differentiation, survival and bone function

The third paragraph about the construction of hypoxic microenvironment using microfluidic platform should better organized. The authors should better explain the utility of this system as an in vitro platform to mimic the bone hypoxic microenvironment.

Authors should not only talk about the materials and the construction of the system but also underline the attention on how these systems can be applied as a model for the study of different bone-diseases

Authors could also add a paragraph about advantages and disadvantages of these platforms compared to other 3D biologically models able to mimic the hypoxic microenvironment (Maansorifar A. et al. https://doi.org/10.1002/adfm.202006796; Liverani C. et al. https://doi.org/10.1038/s41598-019-48701-4)

Reviewer 2 Report

The reviewer would like to appreciate the authors for their efforts. The review was written with good number references and it indeed holds a genuine prospect to explore in future. Thereby, the reviewer recommends a minor inclusion for betterment of the review. 

- Please add a graphical representation of the review exhibiting how the hypoxic condition in the bone marrow initiates. (eg., bone image > magnify a section where the author can detail the hypoxia and report). The representation must be detailed one (highly recommended)

- Try to make the Figure 1 images more interactive and  clear

Reviewer 3 Report

In this review by Li et al, the authors perform a comprehensive review of normal bone physiology and how hypoxia plays into various bone processes. They then move onto to discuss current in vitro bone models including bone on a chip platforms. The figures are nicely done and description of the current models in section 3 is comprehensive. However, This review overall requires more focus on the various in vitro platforms for study of bone physiology in the hypoxic environment. The discussion of the physiology of bone turnover is overall too long and dense and takes up a substantial portion of the review until the various methods for generating models of the hypoxic microenvironment are discussed (section 3). Would recommend  condensing significantly and referring to other reviews to allow the reader to focus more attention on the microfluidics platforms which should be the main focus of the review.

Other smaller comments are as follows:

Line 42- change “film” to review

Line 44-switch “share our vision for the future” with “report the main opportunities and key challenges we have found in this field”

Line 51- change “have matured and promote” to “have matured to promote”

Lines 52-55 This sentence is a combination of three sentences…Please edit this sentence to break it down into smaller sentences.

Line 68- this is an incomplete sentence

Line 70-Range should have lower number listed first

Line 75- endosteal should be endosteum

Line 80- of talus?

Line 99- introduce the HIF abbreviation here

Line 102- delete the first alpha

Line 106- “in the oxygen dependent domain (long)” Unclear what this means

Line 166-orthodontics is not the appropriate term here

Line 188-generates “wrong” signals- please revise

Line 207-this is an incomplete sentence

Line 227- change transformation to transition

Line 413- please add units to the 100

Line 473-475- This is an incomplete sentence

Line 517- Replace “involved in” with “studied by” and change “at present” with “at the present time”

Round 2

Reviewer 1 Report

Authors have accomplished most of the comments arised in the previous revision. Therefore, the manuscript has been ameliorate so that it can be considered for publication in the journal.

Author Response

Thank you very much for your comments. The English language and style have been carefully checked. All of your suggestions are very important and have important guiding significance for my thesis writing and scientific research work. I hope to learn more from you.

Reviewer 3 Report

The authors have addressed the main critiques among the reviewers and this manuscript is much improved in both readibility and content. 

Author Response

Thank you very much for your comments. The English language and style have been carefully checked. All of your comments and suggestions are very important for the manuscript writing. We hope to learn more from you.